# A Generalizable CNN Enhancement for Robust Mammogram Classification Conference Submissions

## Abstract

Deep convolutional neural networks (CNNs) have demonstrated outstanding capabilities in analyzing mammogram images. However, their overall performance is frequently hampered by class imbalance and the inherent complexity of mammogram images. In this work, we propose a novel generalizable enhancement that can be seamlessly integrated into any CNN architecture aimed at improving classification outcomes for mammographic images. In order to rigorously evaluate its effectiveness, the proposed approach was applied to six commonly used CNN models and assessed on an imbalanced multiclass mammogram images dataset. The experimental results showed consistent and significant improvements in all key performance metrics, such as accuracy, precision, recall, F1-score, Precision-Recall Curve (PRC), and Area Under the Curve (AUC), which highlights the robustness and adaptability of our method. This enhancement provides a generalizable strategy for strengthening CNN-based mammogram classification systems, thereby promoting more reliable computer-aided diagnosis in breast cancer screening.

## 1 Introduction

Breast cancer remains a significant worldwide health concern, afflicting women in all corners of the globe. In fact, according to the International Agency for Research on Cancer (IARC), Globally, breast cancer is the most common cancer type among women and the second most common cancer type overall Kim et al. (2025). Certainly, the reduction of mortality rate and the increase of the chances of cure are possible only if this pathology is detected at the early stages, which the early screening through mammogram considers very important to improve the outcomes of breast cancer detection and survival. Deep learning has made significant progress with the introduction of convolutional neural networks (CNNs) in image classification. Their considerable success has had a transformative effect on the detection of breast cancer. Deep convolutional neural networks have achieved impressive performance in the automatic detection and classification of soft tissue opacities in mammograms, reaching a level comparable to that of radiologists in some cases. Indeed, a recent study by Calisto et al. (2022) indicated that the BI-RADS-based DCNN achieved high accuracy, although with slightly lower sensitivity compared to radiologists.

Enhancing the performance of CNN is an ongoing journey. Hardware acceleration, model interpretability, ensemble learning, and advancements in neural architecture search are areas continuously evolving. The quest for higher performance in CNN is driven by the need for more accurate, efficient, and robust models, which means better mammogram images classification (breast cancer diagnosis). However, their performance often depends on the choice of architecture and is hampered by imbalanced datasets and limited generalization. To surmount this limitation, there is an acute need for improvement strategies that not only offer better classification performance, but can also be generalized to various CNN architectures. Most existing improvement methods are closely tied to a specific model, which reduces their applicability and limits wider adoption in clinical settings. In this work, we introduce a generalizable enhancement approach for CNN-based mammogram classification. To demonstrate the method's adaptability and performance regardless of the model selected, it is assessed across multiple CNN architectures namely RegNet, ResNet, MobileNet, Inception,NasNetLarge, and NasNetMobile. Our findings show that the proposed im-

provement to CNNs enhances key metrics such as Accuracy, Precision, Recall, F1-score, AUC, and PRC score across all tested CNN models, highlighting their capabilities in enhancing CNN potential in analyzing mammograms. The main contributions of this paper are:

- We introduce a novel enhancement approach that can be seamlessly integrated into any CNN architecture for mammogram classification.
- We tested the proposed improvement method across several CNN models, demonstrating its adaptability and robustness.
- We validate the proposed approach on a multi-class dataset featuring an imbalance in the number of images per class, demonstrating its ability to maintain performance across underrepresented categories.
- We present a detailed empirical evaluation to demonstrate continuous improvements in performance metrics, confirming the practical relevance of this approach.

The rest of the paper consists of the following sections: Section 2 reviews related work on CNN-based mammogram classification and the improvement techniques of CNN. Section 3 describes the proposed approach in detail. Section 4 introduces the experimental setup and the used dataset. Section 5 presents and discusses the results, and Section 6 concludes the paper with perspectives for future research.

## 2    RELATED WORKS

Deep convolutional neural networks (CNNs) have enabled significant advances in the automated detection of breast cancer on mammograms. Early successes have taken advantage of architectures such as VGG16 and ResNet50, illustrating the capabilities of CNNs for mammogram image classification. However, these models have often encountered difficulties in terms of generalisation and sensitivity to class imbalance.

Transfer learning has become an essential pillar of medical image analysis, enabling models to utilise pre-trained features from large datasets. Recent studies have implemented transfer learning to mammogram classification, reaching remarkable improvements. For instance, Twum et al. (2025) investigated the potential of transfer learning with models such as InceptionV3, MobileNetV2, and ResNet50, assessing their efficiency in feature extraction and fine-tuning applications. Similarly, Nasir et al. (2025) presented a CNN framework for breast cancer detection, focusing on the capabilities of deep learning models to realize high accuracy without the need for intensive manual feature engineering. However, the study was limited by the size and diversity of the dataset, which may limit the robustness of the reported results.

Beyond transfer learning, ensemble learning has appeared as a complementary approach to enhance the performance of CNNs. As an example, Shah et al. (2024) proposed a deep-learning ensemble CNN model that combines EfficientNet, AlexNet, ResNet, and DenseNet for breast cancer detection. The proposed approach showed excellent performance on the evaluated dataset. Specifically, the model achieved an accuracy of 94.6%, underscoring its effectiveness in correctly classifying mammograms. But performance dropped when performing tests on external datasets, revealing challenges with cross-dataset robustness. Recently, Islam et al. (2024) introduced an Ensemble Deep Convolutional Neural Network (EDCNN) combined with a U-Net framework for breast cancer detection and segmentation using ultrasound images. Their approach integrated MobileNet and Xception to leverage complementary feature extraction capabilities. The proposed EDCNN achieved an accuracy of 87.82%, surpassing several transfer learning models and the Vision Transformer baseline. Whilst the results demonstrated clear improvements in classification and segmentation tasks, the study was limited by its reliance on grayscale ultrasound images, which may constrain the model's generalizability to other imaging modalities and more heterogeneous datasets.

To further improve accuracy, ensemble strategies have emerged. Hence, Wu et al. (2023) combined bagging and stacking techniques, achieving 98.84% accuracy in binary classification tasks. While Ganaie et al. (2022) presented a survey on deep ensembles, highlighting their ability to increase robustness. The drawbacks of these methods, however, are often high computational complexity and increased memory usage, which limits their deployment in real-time clinical environments or those with limited resources. Furthermore, as demonstrated by Abe et al. (2024), diversity in lightweight neural network ensembles improves generalization, while in high-capacity models, excessive diversity can induce instability and overfitting.

Data augmentation was also used to handle limited datasets of mammograms. However, conventional techniques, including rotation, flipping, cropping, and density adjustment, have been extensively used to further diversify training and improve model robustness, as in studies by Arevalo et al. (2016), Asifullah Khan (2020), and Shen et al. (2019). Nevertheless, these basic transformations often fail to capture the full complexity of the true changes in mammograms. Recent research has investigated more advanced enhancement approaches to overcome these limitations. Specifically, Sutjiadi et al. (2025) used the diffusion model to generate synthetic mammographic images, which improved the performance of the EfficientNetV2L model in breast cancer screening. Likewise, Jiménez-Gaona et al. (2024) took advantage of generative adversarial networks (GANs) to augment training data, resulting in significant gains in classification. Nguyen et al. (2023) proposed domain-adaptive enhancement strategies that take into account the distribution of breast density and disturbed markers, thus enhancing the detection of abnormal cases. In addition, Panambur et al. (2025) presented an attention-guided erasing (AGE) approach, which focuses on clinically relevant areas. Collectively, these developed enhancement techniques demonstrate that carefully designed synthetic and domain-aware transformations can substantially improve the robustness and generalization of deep learning models in mammogram analysis.

Although prior works often report accuracies above 95%, many are limited to binary classification tasks or small datasets. These settings fail to capture the complexity of real-world clinical environments, where class distributions are imbalanced and image quality varies across devices. Recent studies Litjens et al. (2021), Yamashita et al. (2021) highlight that CNNs trained under such conditions tend to overfit and struggle with generalization across heterogeneous datasets. In summary, while CNNs, ensemble methods, and data augmentation strategies have demonstrated potential in mammogram classification, their effectiveness is often constrained by computational cost, dataset limitations, and generalization challenges, particularly in scenarios involving imbalanced datasets and multi-class classification based on the BI-RADS lexicon. Many existing models struggle to accurately detect rare categories, such as high-risk BI-RADS classes, leading to biased predictions and reduced clinical applicability. Our proposed approach addresses these gaps by introducing a generalizable CNN enhancement that improves robustness and accuracy across all classes while maintaining computational efficiency. By explicitly considering multi-class BI-RADS annotations and handling class imbalance, our method provides a practical pathway toward real-world deployment in breast cancer screening systems.

## 3 PROPOSED APPROACH

Architectural innovations like ResNet, MobileNet, Inception, NasNetLarge, among others, have significantly advanced CNN capabilities. These architectures are designed to address issues like vanishing gradients and overfitting, and to enable models to learn complex features more effectively. Choosing the most suitable architecture for your problem is the first step towards performance improvement. The weights in a CNN play a critical role in allowing the network to learn and generate precise predictions from input data. They are at the heart of the model's learning and decision-making process. CNN weights are typically used to extract significant features from input data, adjust connections between neurons, control model complexity, and optimize performance. Weight learning is the process whereby the CNN acquires the capacity to recognize and interpret complex information present in input data, which is essential for its success in various computer vision tasks. These weights are calculated using a training process that aims to find the optimal values of weights that enable the network to perform accurate predictions on a set of training data, which means that prediction accuracy depends directly on the optimal weights. That is why, in the proposed approach, we focused on optimizing the weights so that CNN achieves good results in classifying mammograms.

However, during the training process, the weight adjustment process essentially consists of minimizing the loss function. In this context, CNN's primary objective is to find weight values that minimize this loss function. To reach this outcome, the model employs an optimization algorithm, such as stochastic gradient descent (SGD) or more advanced variants like Adam, Adamax, and others. The optimization algorithm operates by adjusting the weights of the model based on the gradients calculated through the backpropagation algorithm. The gradients indicate the direction in which the weights should be adjusted to reduce the loss. Subsequently, the optimization algorithm iteratively updates the weights using these gradients. This weight adjustment process is repeated over the entire training dataset for several epochs (complete iterations through the dataset) or until

the model's performance no longer improves. Over epochs, the CNN gradually refines its weights, allowing it to better adapt to the features of the training data. Ultimately, in the evaluation phase, the weights used are the final weights that have been learned during the training phase. These final weights represent the optimized parameters that the CNN has learned for the specific task, and the model uses them to make predictions on new, previously unseen data to evaluate its performance and generalization ability. Using the fixed weights during evaluation ensures that the model retains the knowledge acquired during training and applies it to make predictions on new data in a consistent way. This phase is essential for assessing the model's ability to generalize to examples not seen during training, which is a crucial indicator of its actual performance.

Given the significance of weight and its vital role in the performance of the CNN network, our contribution lies in utilizing the best weights with the lowest loss during the evaluation phase, as achieved during the training phase. As previously mentioned, the weights used during the evaluation phase are generally the most recent ones, irrespective of whether they yield optimal performance. For this reason, we advocate utilizing the weights that yield the best performance during the training phase, with the loss function serving as the evaluation metric. While learning a CNN model, the weights are iteratively updated in order to minimize the loss function. However, the model's performance can vary from one iteration to another, and it may not always converge to the best solution. This is why we aim to monitor the model's performance over the course of training and record the weights that perform best for a specific measure, namely loss. By specifying loss as the measure to be monitored, model weights will be saved solely if the monitored measure exhibits improvement on the previous best value. This approach ensures that the weights associated with the highest model performance are consistently saved while ignoring checkpoints that show no enhancement in model performance. Therefore in the evaluation phase, the model will be evaluated with the saved weights using performance metrics such as accuracy, precision, recall, F1-score, AUC, and PRC on the test data.

---

**Algorithm 1** Intial Algorithm for CNN Mammogram Classification

---

1: **Input:** CNN_Model, train_Dataset, test_Dataset, nb_Epoch
2: **Output:** metrics = {Accuracy, Precision, Recall, F1, AUC, PRC}
3: **for** i = 1 to nb_Epoch **do**
4:     weights ← train_Model(CNN_Model, train_Dataset)     ▷ Train model and get weights
5:
6: **end for**
7: load_Weights(CNN_Model, weights)     ▷ Load model weights at the last epoch
8: metrics ← evaluate_Model(CNN_Model, test_Dataset)     ▷ Evaluate on test set using weights at the last epoch

---

**Algorithm 2** Proposed Algorithm for CNN Mammogram Classification

---

1: **Input:** CNN_Model, train_Dataset, test_Dataset, nb_Epoch
2: **Output:** metrics = {Accuracy, Precision, Recall, F1, AUC, PRC}
3: best_Loss ← ∞     ▷ Initialize best loss as infinity
4: best_Weights ← Null
5: **for** i = 1 to nb_Epoch **do**
6:     loss ← train_Model(CNN_Model, train_Dataset)     ▷ Train model and get loss
7:     **if** loss < best_Loss **then**
8:         best_Loss ← loss
9:         best_Weights ← save_Weights(CNN_Model)     ▷ Save best weights
10:     **end if**
11: **end for**
12: load_Weights(CNN_Model, best_Weights)     ▷ Load best weights into model
13: metrics ← evaluate_Model(CNN_Model, test_Dataset) ▷ Evaluate on test set using best weights

---

Table 1: Parameters used in training CNN models

| Parameter | Values |
|---|---|
| Learning rate | 0.001 |
| mini-batch size | 32 |
| Optimizer | Adamax |
| Epochs | 200 |
| Early stopping | 5 |

## 4 EXPERIMENTS

### 4.1 EXPERIMENTS DESCRIPTION

Our study applied transfer learning to six CNN models, including RegNet [Xu et al. (2021)], ResNet152 [He et al. (2016)], MobileNetV3 [Howard et al. (2019)], InceptionV3 [Szegedy et al. (2016)], NasNetLarge [Zoph et al. (2018a)] and NasNetMobile224 [Zoph et al. (2018b)], used for breast cancer diagnosis and classification based on BI-RADS lexicon. The CNN models are used as a feature extractor while keeping their initial architecture. In this process, the lower layers of the feature extractor portion are frozen, while the original fully connected, softmax and classification output layers are eliminated and substituted with a new set. This new set has an output size of 6, signifying the multi-classification of breast cancer based on BI-RADS lexicon ( BI-RADS 0, BI-RADS 1,BI-RADS 2, BI-RADS 3, BI-RADS 4, and BI-RADS 5).

As part of an initial experiment, we didn't seek to improve the CNN models or adjust the model weights throughout the feature learning sections. We therefore simply applied transfer learning to the CNN models on our small dataset to evaluate the performance of each CNN model in the specific task of breast cancer diagnosis in a multi-class context (6 classes).

Then, as part of a second experiment, we applied our approach, which involves adjusting the weights during the testing phase as explained in Section 3. In the testing phase, we used the weights that yielded the best performance during the training phase, rather than employing the final weights learned during the training phase. This stage aims to demonstrate the performance of our proposed approach across several CNN architectures, including RegNet, ResNet, MobileNet, Inception,NasNetLarge, and NasNetMobile, which are widely recognized as high-performing deep models for classification tasks. The approach was applied to six different CNN models to show that it consistently improves performance, highlighting its generalizability across diverse architectures.

In the two experiments for the training of the six CNN models, we worked with the Adamax optimizer, and interestingly, there was no discernible difference in the achieved performance during the initial experiments when compared to the SGD and Adam optimizers. Each CNN model was trained in 200 epochs before being put to the test stage. As the used mini-batch size is equal to 32 and the learning rate is equal to 0.001, as shown in Table 1. Additionally, using early stopping with a patience value of 5 ensures that the training process will come to an end if there is a prolonged period without noticeable performance progress. This strategy saves computational resources while reducing the danger of overfitting. The hyperparameter values typically demonstrate a well-calibrated configuration aimed at striking a harmonious balance between effective learning from the given data and minimizing the risk of overfitting. It's essential to take into account that the setting of hyperparameters is frequently an iterative and exploratory process. The effectiveness of these choices is generally confirmed by tests carried out on a selected dataset. For each experiment, we tested the CNN model 5 times in order to obtain a new training set and a new test set for each evaluation, using the random split technique.

### 4.2 DATASET DESCRIPTION

The dataset used in these experiments is a collection of mammogram images that have been acquired from different sources. The first source is the King Abdulaziz University Mammogram Dataset that consists of 5 classes, which are BI-RADS 1, BI-RADS 2, BI-RADS 3, BI-RADS 4, and BI-RADS 5 [Alsolami et al. (2021)]. These images were gathered from the Sheikh Mohammed Hussein Al-Amoudi Center of Excellence in Breast Cancer at King Abdulaziz University in Jeddah,

Saudi Arabia, spanning the period from April 2019 to March 2020. The total number of images is 5662, with different image sizes encompassing two distinct views, namely CC (Cranio-Caudal) and MLO (Medio-Lateral Oblique), for both the right and left breasts. The second source is the Digital Database for Screening Mammography (DDSM) [Heath et al. (2001)] which comprises 6 classes of BI-RADS classification ranging from 0 to 5th class. As mentioned above, we combined mammogram images from two datasets for the evaluation of our approach because the King Abdulaziz University mammogram dataset does not include all the classes; only BI-RADS 1, BI-RADS 3, BI-RADS 4, and BI-RADS 5 are public. To address this, we added the BI-RADS 0 and BIRADS 2 classes from the DDSM dataset, resulting in a final dataset comprising 2,881 images distributed as follows:

- BI-RADS 0: 225.
- BI-RADS 1: 1865.
- BI-RADS 2: 278.
- BI-RADS 3: 387.
- BI-RADS 4: 102.
- BI-RADS 5: 24.

From this distribution, we can see that the used dataset is imbalanced, which is often the case in medical datasets, especially for rare conditions or diseases. Managing these imbalances is essential to ensure that deep convolutional neural networks can learn efficiently and produce reliable results. We can solve the imbalance problem through many techniques, such as data augmentation. Since the use of imbalanced data ensures that CAD systems are trained on datasets that reflect the actual distribution of cases, which makes them more applicable to real-life scenarios. Consequently, the dataset will be used in this evaluation exactly as it is, without any changes.

Therefore, some preprocessing steps could be applied before the experiments to prepare the data as soon as a good data preparation guarantees that the model receives high-quality data, which facilitates its learning and enables it to produce accurate and consistent results during the classification phase. There are several crucial steps in this procedure. We first resized all mammogram images in the dataset to 331x331 pixels to have uniform dimensions, making them compatible with the CNN model architecture. By resizing images to a reasonable, standardized size, computing costs in both memory and processing power are significantly reduced, enabling these models to be trained and deployed on different hardware. Then, all grayscale mammogram images are converted to RGB so that the CNN models can process them because the CNN architectures employed in this evaluation just require RGB images. Finally, the dataset is randomly partitioned, allocating 80% of the mammogram images to training and reserving the remaining 20% for testing. In the experiments, the training set included 2304 mammogram images, while the test set included the remaining 577 mammogram images. This randomization reduces the risk of bias in the evaluation. By randomly mixing and distributing the data, it is ensured that each set involves a variety of samples from all the categories in the dataset, in each test. As a result, applying random splitting to imbalanced datasets is an essential deep learning technique that preserves data integrity, guarantees objective, strong, and reliable model evaluation, and aids in the creation of models that can successfully address class imbalance problems in real-world applications.

### 4.3 ASSESSMENT METRICS

To provide a comprehensive evaluation of the proposed approach, several quantitative assessment metrics were employed. These metrics capture not only overall classification performance but also the clinical relevance of the results:

- Accuracy: Measures the overall proportion of correctly graded cases.
- Precision: is the accuracy of positive predictions, reflecting dependability in detecting abnormal cases.
- Recall (Sensitivity): Measures the ability to correctly identify all relevant instances of a particular class, which is critical in medical diagnosis to reduce false negatives.
- F1-score: is commonly used to assess overall model performance, particularly in situations where datasets are imbalanced.

- Area Under the ROC Curve (AUC): Represents the model's ability to distinguish between classes across different thresholds.

- Precision-Recall Curve (PRC): Offers a more informative evaluation than AUC in imbalanced datasets, highlighting the trade-off between precision and recall.

By combining these metrics, the evaluation ensures that the proposed method is assessed from multiple perspectives, addressing both statistical performance and clinical significance.

## 5 RESULTS AND DISCUSSION

To further demonstrate the effectiveness of the proposed approach during the training and testing phases, we have summarized the quantitative evaluation metrics including accuracy, precision, recall, AUC and PRC, and F1-score in table 2. The overall metrics were calculated by averaging the values across all five test sets in our experiments. As well, this table provides a clear before-and-after comparison, showcasing the impact of our approach on enhancing model performance across six CNN architectures. The striking difference between the results of all CNN models without improvement and those with the proposed enhancement is truly remarkable. Regarding accuracy performance, all enhanced models reached higher values compared to their baselines.

It is worth noting that InceptionV3 achieved the largest accuracy improvement, rising from 79.12% to 94.41%, while ResNet152 rose from 74.96% to 86.73%. RegNet also exhibited a considerable increase, reaching 89%, compared to 94.65% in its baseline.Besides accuracy improvements, the precision and recall metrics underscore the effectiveness of the proposed method. Precision recorded a significant increase for MobileNet rising from 83.48% to 94.04% and InceptionV3 up from 83.76% to 94.49%. Recall, a crucial indicator for medical applications where diagnostic errors are highly undesirable, improved substantially for InceptionV3 from 73.72% to 94.27%, and for ResNet152, from 68.92% to 82.46%. In addition the F1-score values showed consistent improvements, highlighting a better balance between precision and recall. For example, NasNetLarge strengthened from 0.83 to 0.88, whereas InceptionV3 rose from 0.73 to 0.93, representing an increase of more than 0.2. The proposed models reached strong performance in terms of AUC, with most of the enhanced models achieving values above 0.95. Similarly, PRC values demonstrated the robustness of the improved models against class imbalance, with RegNet and NasNetMobile224 both attaining 0.98. These results confirm that the enhanced CNNs illustrate a reliable distinction between positive and negative cases. The comparative results reveal three key findings:

- Generalizability across architectures: The proposed enhancement consistently optimises performance in lightweight networks (MobileNet) as well as deeper architectures (InceptionV3, ResNet152), revealing that the proposed method is not architecture-dependent.

- Clinical relevance: Given that the accuracy rate reaches 15% with InceptionV3, these improvements are particularly valuable for breast cancer diagnosis, since they decrease the risk of missed diagnoses.

- Robustness: Both high AUC and PRC values confirm that the enhanced CNN models maintain reliable discrimination even in the presence of imbalanced datasets.

Nevertheless, some basic models, such as ResNet152, initially had lower recall and F1-score values, emphasising the importance of the proposed improvement. Besides enhancing individual metrics, the method also guarantees more consistent performance across diverse CNN architectures. Overall, the proposed approach provides a generalisable improvement to CNN-based mammogram classification, enhancing both discriminative power and reliability, which is crucial for clinical applicability.

Figure 1 illustrates the accuracy results for each CNN model across five independent test runs. Each CNN model including RegNet, ResNet152, InceptionV3, NasNetLarge, NasNetMobile224, and MobileNetV3 was assessed five times on different test dataset to guarantee robustness and assess variability. Both the baseline and the improved versions of all models are shown, allowing a direct comparison of effectiveness gains. On the whole, the enhanced models consistently exceeded their corresponding baselines in all tests. In particular, InceptionV3 and RegNet demonstrated the most significant accuracy improvements. RegNet achieved baseline accuracy between 75.6% and 94.1% across the five tests. After implementing the proposed improvement approach, the enhanced RegNet model consistently maintained accuracy ranging from 94.4% to 95.1%, demonstrating both

Table 2: The CNN models results before and after applying proposed approach

| CNN model | Overall accuracy | Overall precision | Overall recall | Overall F1-score | Overall AUC | Overall PRC |
|---|---|---|---|---|---|---|
| RegNet | 89% | 89.59% | 88.41% | 0.88 | 0.95 | 0.94 |
| **Improved RegNet** | 94.65% | 94.8% | 94.27% | 0.94 | 0.97 | 0.98 |
| ResNet152 | 74.96% | 85.18% | 68.92% | 0.68 | 0.88 | 0.86 |
| **Improved ResNet152** | 86.73% | 86.73% | 82.46% | 0.82 | 0.94 | 0.94 |
| InceptionV3 | 79.12% | 83.76% | 73.72% | 0.73 | 0.87 | 0.82 |
| **Improved InceptionV3** | 94.41'% | 94.49% | 94.27% | 0.93 | 0.96 | 0.97 |
| NasNetLarge | 85.04% | 89.45% | 84.13% | 0.83 | 0.92 | 0.87 |
| **Improved NasNetLarge** | 92.22% | 93.11% | 89.41% | 0.88 | 0.95 | 0.95 |
| NasNetMobile224 | 92.19% | 93.21% | 91.54% | 0.91 | 0.96 | 0.94 |
| **Improved NasNetMobile224** | 95.47% | 96.43% | 94.66% | 0.95 | 0.98 | 0.98 |
| MobileNetV3 | 80.61% | 83.48% | 78.44% | 0.79 | 0.88 | 0.83 |
| **Improved MobileNetV3** | 89.85'% | 94.04% | 85.57% | 0.87 | 0.96 | 0.95 |

a substantial increase in performance and greater stability. Further, InceptionV3 exhibited baseline variability in accuracy values, with a notable decline to 79.5% in subsequent tests after reaching a peak of 94.1%, implying overfitting issues. In contrast, the improved model achieved accuracies above 93%, highlighting increased robustness and generalisation. Over the six models, the proposed approach has consistently improved both accuracy and stability. Models subject to high levels of variability or slower convergence, notably ResNet152, InceptionV3, and MobileNetV3, were the most beneficial, with a significant reduction in performance fluctuations. High-performing models, in particular the RegNet and NasNet variants, showed more modest but consistent improvements, confirming that the approach is generalisable to different CNN architectures. The results indicate that the proposed improvement approach not only enhances performance, but also reduces the risk of overfitting, as shown by smoother accuracy curves in the improved models. This robustness is particularly important in medical imaging tasks, where classification errors can have critical consequences.

## 6   CONCLUSION

In this paper, we proposed a generalisable enhancement approach for deep convolutional neural networks designed for mammogram images classification. This method is specifically designed to address the challenge of class imbalance in mammographic datasets, which is critical for accurate breast cancer detection and BI-RADS categorization. The approach was evaluated on six state-of-the-art CNN architectures, namely RegNet, ResNet152, InceptionV3, NasNetLarge, NasNetMobile224, and MobileNetV3. The experimental results proved that the improved models consistently outperformed their baseline counterparts in terms of accuracy, precision, recall, F1 score, AUC, and PRC. The improvements were particularly impressive in terms of recall and F1 score, which are key indicators for medical diagnosis, guaranteeing a significant reduction in false negatives and a greater balance between precision and sensitivity. The proposed approach has proven efficient for both shallow and deep architectures, thus confirming its generalisation and robustness, even in the case of class imbalance typical of mammogram images datasets. Overall, the proposed enhancement offers a reliable and adaptable strategy for improving the performance of convolutional neural networks in the imbalanced multi-class classification of mammographic images, with potential applications in computer-aided diagnosis systems. Our future work will involve extending this approach to multimodal medical imaging data and optimising model interpretability to facilitate clinical decision-making.

## REFERENCES

Taiga Abe, E. Kelly Buchanan, Geoff Pleiss, and John Patrick Cunningham. Pathologies of predictive diversity in deep ensembles. *Transactions on Machine Learning Research*, 2024. ISSN

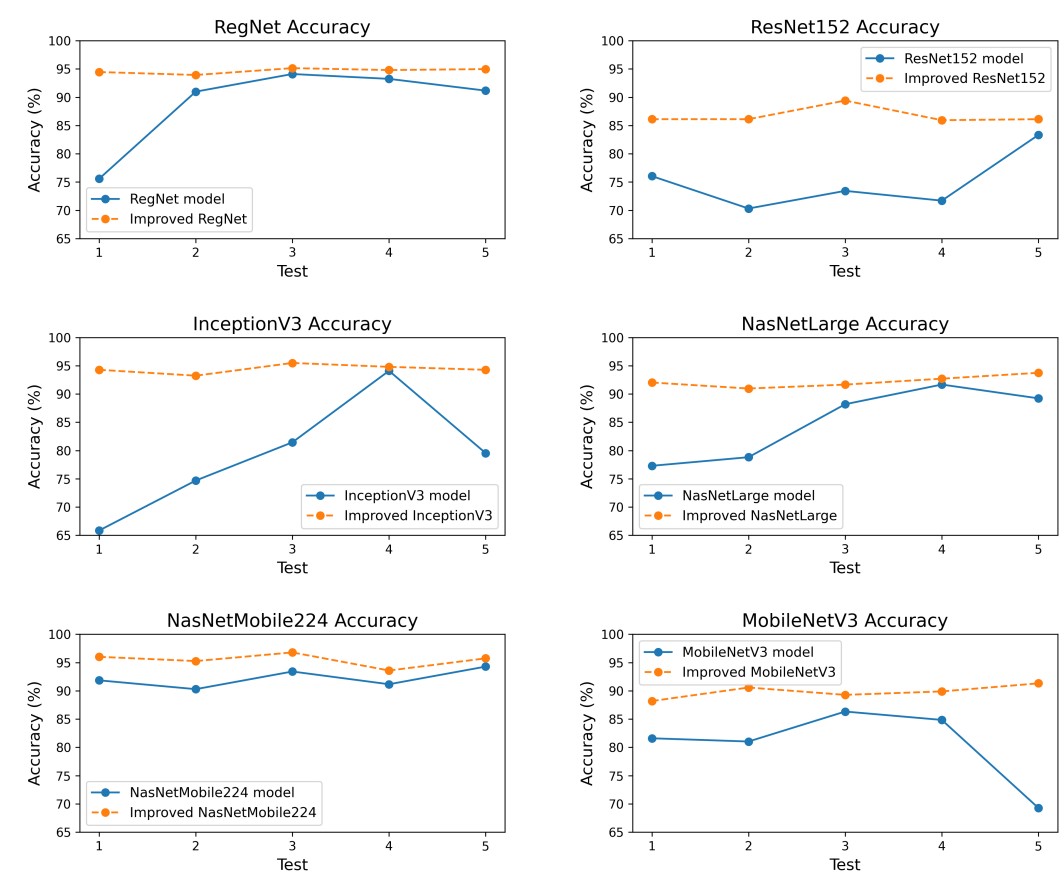

Figure 1: CNN Models Accuracy Improvement.

2835-8856.

Asmaa S. Alsolami, Wafaa Shalash, Wafaa Alsaggaf, Sawsan Ashoor, Haneen Refaat, and Mohammed Elmogy. King abdulaziz university breast cancer mammogram dataset (kau-bcmd). *Data*, 6(11):1, 2021.

John Arevalo, Fabio A. González, Raúl Ramos-Pollán, José Luís Oliveira, and Miguel Ángel Guevara-López. Representation learning for mammography mass lesion classification with convolutional neural networks. *Computer methods and programs in biomedicine*, 127:248–57, 2016.

Umme Zahoora Aqsa Saeed Qureshi Asifullah Khan, Anabia Sohail. A survey of the recent architectures of deep convolutional neural networks. *Artificial Intelligence Review*, 53:1–87, 2020.

A. Calisto, C. R. Cardoso, J. R. C. Silva, and A. C. Campilho. Bi-rads-based classification of mammographic soft tissue opacities using a deep convolutional neural network. *Diagnostics*, 12 (7):1564, 2022. doi: 10.3390/diagnostics12071564.

Mohammad Arif Ganaie, Minghui Huand, A. K. Malik, Muhammad Tanveer, and Ponnuthurai Nagaratnam Suganthan. Ensemble deep learning: A review. *Engineering Applications of Artificial Intelligence*, 115, 2022.

Kaiming He, Xiangyu Zhang, Shaoqing Ren, and Jian Sun. Deep residual learning for image recognition. In *Proceedings of the IEEE Conference on Computer Vision and Pattern Recognition (CVPR)*, pp. 770–778, 2016. doi: 10.1109/CVPR.2016.90.

Michael Heath, Kevin Bowyer, Daniel Kopans, Richard Moore, and W. Philip Kegelmeyer. The digital database for screening mammography. In *Proceedings of the Fifth International Workshop*

*on Digital Mammography*, pp. 212–218. Medical Physics Publishing, 2001. ISBN 1-930524-00-5.

Andrew Howard, Mark Sandler, Grace Chu, Liang-Chieh Chen, Bo Chen, Mingxing Tan, Weijun Wang, Yukun Zhu, Ruoming Pang, Vijay Vasudevan, Quoc V Le, and Hartwig Adam. Searching for mobilenetv3. *arXiv preprint arXiv:1905.02244*, 2019.

Md Rakibul Islam, Md Mahbubur Rahman, Md Shahin Ali, Abdullah Al Nomaan Nafi, Md Shahariar Alam, Tapan Kumar Godder, Md Sipon Miah, and Md Khairul Islam. Enhancing breast cancer segmentation and classification: An ensemble deep convolutional neural network and u-net approach on ultrasound images. *Machine Learning with Applications*, 16, 2024.

Yuliana Jiménez-Gaona, Diana Carrión-Figueroa, Vasudevan Lakshminarayanan, and María José Rodríguez-Álvarez. Gan-based data augmentation to improve breast ultrasound and mammography mass classification. *Biomedical Signal Processing and Control*, 94, 2024. ISSN 1746-8094.

J. Kim, A. Harper, V. McCormack, et al. Global patterns and trends in breast cancer incidence and mortality across 185 countries. *Nature Medicine*, 31:1154–1162, 2025. doi: 10.1038/s41591-025-03502-3. URL https://doi.org/10.1038/s41591-025-03502-3.

G. Litjens, T. Kooi, B. E. Bejnordi, A. A. A. Setio, F. Ciompi, M. Ghafoorian, J. A. W. M. van der Laak, B. van Ginneken, and C. I. Sánchez. A survey on deep learning in medical image analysis. *Medical Image Analysis*, 42:60–88, 2021.

Faizan Nasir, Shanur Rahman, and Nazim Nasir. Breast cancer detection using convolutional neural networks : A deep learning-based approach. *Cureus*, 17(5):e83421, 2025.

H. T. Nguyen, T. B. Lam, Q. T. D. Tran, M. T. Nguyen, D. T. Chung, and V. Q. Dinh. Gan-based data augmentation to improve breast ultrasound and mammography mass classification. *arXiv preprint arXiv:2306.06893*, 2023.

A. B. Panambur, S. Bhat, H. Yu, P. Madhu, S. Bayer, and A. Maier. Attention-guided erasing for enhanced transfer learning in breast abnormality classification. *International Journal of Computer Assisted Radiology and Surgery*, 19(20(3)):433–440, 2025.

Dilawar Shah, Mohammad Asmat Ullah Khan, Mohammad Abrar, and Muhammad Tahir. Optimizing breast cancer detection with an ensemble deep learning approach. *International Journal of Intelligent Systems*, 2024.

Li Shen, R. Margolies Laurie, Rothstein Joseph, H., Eugene Fluder, Russell McBride, and Weiva Sieh. Deep learning to improve breast cancer detection on screening mammography. *Sci Rep*, 9(1), 2019.

Raymond Sutjiadi, Siti Sendari, Heru Wahyu Herwanto, and Yosi Kristian. Leveraging synthetic mammograms to enhance deep-learning performance for breast cancer classification using efficientnetv2l architecture. *EAI Endorsed Transactions on AI and Robotics /*, 4, 2025.

Christian Szegedy, Vincent Vanhoucke, Sergey Ioffe, Jonathon Shlens, and Zbigniew Wojna. Rethinking the inception architecture for computer vision. In *Proceedings of the IEEE Conference on Computer Vision and Pattern Recognition (CVPR)*, pp. 2818–2826, 2016. doi: 10.1109/CVPR.2016.308.

Frimpong Twum, Charlyne Carol Eyram Ahiable, Stephen Opoku Oppong, Linda Banning, and Kwabena Owusu-Agyemang. Employing transfer learning for breast cancer detection using deep learning models. *PLOS Digit Health*, 4(6):e0000907, 2025.

Peihceng Wu, Runze Ma, and Teoh Teik Toe. Stacking-enhanced bagging ensemble learning for breast cancer classification with cnn. *International Conference on Electronic Engineering (ICEEM)*, 2023.

Jing Xu, Yu Pan, Xinglin Pan, Steven Hoi, Zhang Yi, and Zenglin Xu. Regnet: Self-regulated network for image classification. *arXiv preprint arXiv:2101.00590*, 2021.

R. Yamashita, K. Nishida, T. Kubo, Y. Matsumoto, T. Fukui, R. Yamaguchi, Y. Sato, T. Sato, M. Kubo, and T. Kubo. Deep learning for breast cancer diagnosis using ultrasound images. *Scientific Reports*, 11, 2021.

Barret Zoph, Vijay Vasudevan, Jonathon Shlens, and Quoc V. Le. Learning transferable architectures for scalable image recognition. In *Proceedings of the IEEE Conference on Computer Vision and Pattern Recognition (CVPR)*, pp. 8697–8710, 2018a. doi: 10.1109/CVPR.2018.00907.

Barret Zoph, Vijay Vasudevan, Jonathon Shlens, and Quoc V. Le. Learning transferable architectures for scalable image recognition. In *Proceedings of the IEEE Conference on Computer Vision and Pattern Recognition (CVPR)*, pp. 8697–8710, 2018b. doi: 10.1109/CVPR.2018.00907.

