# OpenReview forum: "A Generalizable CNN Enhancement for Robust Mammogram Classification"
_ICLR.cc/2026/Conference — ICLR 2026 Conference Desk Rejected Submission_

### Official Review · Reviewer_cVAS · 2025-10-27

**Soundness:** 1
**Presentation:** 1
**Contribution:** 1
**Rating:** 0
**Confidence:** 4

**Summary:**

The paper proposes a "weight selection optimization" enhancement strategy applicable to various CNNs, aiming to improve accuracy in multi-class, imbalanced mammography (BI-RADS) classification.  However, the method essentially employs the "minimum loss weight" from the training phase for evaluation during testing, lacking substantive innovation and being supported by insufficient experimental design and evidence.  Therefore, the paper is recommended for rejection.

**Strengths:**

See Summary.

**Weaknesses:**

1.  Lack of innovation: The method proposed in the paper is similar to the common model checkpointing early stopping paradigm, lacking distinctiveness.  It fails to establish a new theoretical framework or introduce novel mechanisms, nor does it provide a comparative analysis with traditional methods.
2.  Lack of comparison with cutting-edge requirements: The experiments in this paper only compare "the same CNN before and after application enhancement" rather than benchmarking against external SOTA methods, failing to provide a comparison with the latest research approaches from top-tier conferences and journals.
3.  Supplementary Related Work: The discussion on related work should be more in-depth, with a suggested addition to the discussion on deep learning-based mammography classification methods.
4.  Lack of Visualization in Experiments: Most experimental data relationships in the paper are not presented with intuitive charts, making it difficult to evaluate the actual improvements and interpretability of the method across different categories and difficult cases.
5.  No ablation experiments: The paper only mentions "randomly splitting the dataset 5 times," which constitutes merely a repetitive validation of data partitioning rather than an ablation study.
6.  Missing Overall Architecture/Flowchart: There is no complete experimental flowchart, making it difficult to fully grasp the core of the paper.  The only figure provided in the paper is the accuracy curve.
7.  Ambiguous Methodology: The lack of a clear definition for "optimal weights" in the paper may lead to the risk of overfitting.
8.  Lack of in-depth analysis of training process details: The paper mentions the use of optimizers and early stopping strategies during training but does not analyze the impact of hyperparameter selection on the results or explore the performance differences of models under various optimizers and learning rate settings.

**Questions:**

See Weakness.

---

### Official Review · Reviewer_7NBu · 2025-10-30

**Soundness:** 1
**Presentation:** 1
**Contribution:** 1
**Rating:** 0
**Confidence:** 4

**Summary:**

This paper presents a generalizable enhancement strategy for convolutional neural networks (CNNs) designed to improve mammogram image classification under class imbalance conditions. The proposed method focuses on optimizing weight selection by saving and using the best-performing weights during evaluation rather than the final epoch weights.

**Strengths:**

The paper follows a scientific structure.

**Weaknesses:**

The proposed enhancement primarily selects the lowest-loss weights during training, which is a standard practice in many fields, and there is no theoretical innovation beyond common weight-saving mechanisms.

**Questions:**

The paper only presents a very common method with no innovation, so I have no further questions.

---

### Official Review · Reviewer_b3Up · 2025-10-31

**Soundness:** 1
**Presentation:** 1
**Contribution:** 1
**Rating:** 0
**Confidence:** 5

**Summary:**

This submission attempts to propose a "generalizable enhancement" for convolutional neural networks (CNNs) to improve mammogram classification, aiming to address challenges like class imbalance and limited generalization of existing CNN-based methods. The core idea of the proposed approach is to monitor the loss function during training and save the "optimal weights" corresponding to the minimum loss, instead of using the weights from the final training epoch as in traditional practices. The method is evaluated on six CNN architectures (RegNet, ResNet152, MobileNetV3, InceptionV3, NasNetLarge, NasNetMobile224) using an imbalanced multiclass mammogram dataset (2,881 images, BI-RADS 0–5). However, the paper suffers from three critical flaws: first, the research objective is ambiguously articulated (e.g., the connection between weight selection and solving class imbalance is not clearly justified); second, the related work section is incomplete (failing to cover key relevant methods with similar ideas); third, the experimental design is insufficient—only comparing the proposed method with the "baseline (unimproved) version" of each CNN model, without any comparison with existing state-of-the-art methods (e.g., transfer learning or ensemble learning approaches discussed in its own related work), leading to weak persuasiveness of the claimed "generalizability" and "effectiveness".

**Strengths:**

No obvious strengths are identified in this submission. The core idea of "saving optimal weights based on loss" is a relatively trivial adjustment to standard CNN training pipelines, and the paper fails to demonstrate its novelty, practical value, or superiority over existing techniques.

**Weaknesses:**

1. **Insufficient Literature Review and Lack of Novelty Acknowledgment**. The proposed approach of monitoring the loss function during training bears strong similarities to curriculum learning paradigms. However, the authors have not conducted adequate literature retrieval—there is no citation of such relevant work, nor any discussion or comparison to distinguish the proposed method from these existing techniques. This omission raises concerns about the novelty of the work and suggests a lack of understanding of the broader research landscape.

2. **Inadequate Experimental Comparison**. While the related work section mentions important techniques for mammogram classification, the experiments only compare the "improved" CNN models (with the proposed weight selection) against their own "unimproved baselines" (using final-epoch weights). There is no quantitative or qualitative comparison with the aforementioned transfer learning, ensemble learning, or other state-of-the-art methods. Without such comparisons, it is impossible to verify whether the proposed method outperforms existing solutions or merely provides a marginal improvement over a naive baseline—greatly undermining the paper’s claim of "enhancing CNN-based mammogram classification systems".

3. **Poor Organization and Unclear Research Objectives**. The paper lacks a coherent structure and clear research focus. The research objective (e.g., "how does selecting optimal weights address class imbalance?") is never explicitly and logically justified—only stating that the dataset is imbalanced but not explaining why the proposed weight selection strategy can mitigate this issue. Overall, the writing requires substantial revision to improve clarity and logical flow.

**Questions:**

1. Regarding the first weakness: Are the authors aware of curriculum learning methods that use loss to guide parameter selection?
2. Regarding the second weakness: Given that the related work discusses transfer learning and ensemble learning as competitive techniques, why were these methods not included in the experimental comparisons?
3. Regarding the third weakness: The paper claims to "address class imbalance", but the connection between "weight selection based on loss" and "solving class imbalance" is unclear. Can the authors clarify this mechanism?

---

### Official Review · Reviewer_x1Tr · 2025-11-02

**Soundness:** 1
**Presentation:** 1
**Contribution:** 1
**Rating:** 0
**Confidence:** 5

**Summary:**

The paper prosposes an enhancement on the CNN archtecture. It uses the best set of weights/model among all the epochs and uses it for testing or evaluation. Experiments are shown on several aarchtecture and on breas cancer mammography datasets.

**Strengths:**

The idea is presented in simple manner. Paper is easy to follow.

**Weaknesses:**

The technical contribution is very limtied (close to none).

The abstract and the introducton even do not mention what the enhancement is.

Selection the best weights among the epochs is not a new or novel idea, its a trivial contribution.

**Questions:**

NA

---

### Note · Program_Chairs · 2026-01-17
**Submission Desk Rejected by Program Chairs**

The following references in this submission do not refer to real documents and/or have major errors in bibliographic information:

 R. Yamashita, K. Nishida, T. Kubo, Y. Matsumoto, T. Fukui, R. Yamaguchi, Y. Sato, T. Sato, M. Kubo, and T. Kubo. Deep learning for breast cancer diagnosis using ultrasound images. Scientific Reports, 11, 2021.